

# Facility level measurement of off-shore oil & gas installations from a small airborne platform: Method development for quantification and source identification of methane emissions

James France[1,2], Prudence Bateson[3], Pamela Dominutti[4], Grant Allen[3], Stephen Andrews[4], Stephane Bauguitte[5], Max Coleman[2], Tom Lachlan-Cope[1], Rebecca Fisher[2], Langwen Huang[3&9], Anna E Jones[1], James Lee[6], David Lowry[2], Joseph Pitt[3&8], Ruth Purvis[6], John Pyle[7], Jacob Shaw[3], Nicola Warwick[7], Alexandra Weiss[1], Shona Wilde[4], Jonathan Witherstone[1], Stuart Young[4].

[1] British Antarctic Survey, Natural Environment Research Council, Cambridge CB3 0ET, UK
[2] Department of Earth Sciences, Royal Holloway, University of London, Egham TW20 0EX, UK
[3] Department of Earth and Environmental Science, University of Manchester, Manchester, M13 9L, UK
[4] Wolfson Atmospheric Chemistry Laboratories, Department of Chemistry, University of York, Heslington, YO10 5DD, UK
[5] FAAM Airborne Laboratory, National Centre for Atmospheric Science, Cranfield, Bedfordshire, MK43 0AL, UK
[6] National Centre for Atmospheric Science, Innovation Way, University of York, York, UK
[7] National Centre for Atmospheric Science, Department of Chemistry, University of Cambridge, Cambridge, UK
[8] School of Marine and Atmospheric Sciences, Stony Brook University, Stony Brook, NY 11974, USA
[9] Departement Mathematik, ETH Zurich, Rämistrasse 101, 8092 Zürich, Switzerland

*Correspondence to*: James L France (james.france@rhul.ac.uk)

**Abstract.**

Emissions of methane ($CH_4$) from offshore oil and gas installations are poorly ground-truthed and quantification relies heavily on the use of emission factors and activity data. As part of the United Nations Climate and Clean Air Coalition (UN CCAC) objective to study and reduce short-lived climate pollutants (SLCP) a Twin Otter aircraft was used to survey $CH_4$ emissions from UK and Dutch offshore oil and gas installations. The aims of the surveys were to i) identify installations that are significant $CH_4$ emitters, ii) separate installation emissions from other emissions using carbon-isotopic fingerprinting and other chemical proxies, iii) estimate $CH_4$ emission rates, and iv) improve flux estimation (and sampling) methodologies for rapid quantification of major gas leaks.

In this paper, we detail the instrument and aircraft set up for two campaigns flown in the springs of 2018 and 2019 over the southern North Sea and describe the developments made in both planning and sampling methodology in order to maximise the quality and value of the data collected. We present example data collected from both campaigns to demonstrate the challenges encountered during offshore surveys, focussing on the complex meteorology of the marine boundary layer, and sampling discrete plumes from an airborne platform. The uncertainties of $CH_4$ flux calculations from measurements under varying boundary layer conditions are considered, as well as recommendations for attribution of sources through either spot sampling for VOCs / $\delta^{13}C_{CH4}$ or using in-situ instrumental data to determine $C_2H_6$-$CH_4$ ratios. A series of recommendations for both planning and measurement techniques for future offshore work within the marine boundary layers are provided.



## 1. Overview

Methane is a potent greenhouse gas in the atmosphere, with a global warming potential 84 times that of carbon dioxide when calculated over a 20-year period (Myhre et al., 2013). Increases in atmospheric $CH_4$ mixing ratios are expected to have major influences on the Earth's climate, and emission mitigation could go some way toward achieving goals laid out in the UNFCCC
Paris Agreement (Nisbet et al., 2019).

Offshore oil and gas fields make up ~28% of the total global oil and gas production worldwide and are expected to be significant sources of $CH_4$ to the atmosphere, given that 22% of global $CH_4$ emissions are estimated to be from the oil and gas (O&G) sector (Saunois et al., 2016). Some emissions arise from routine operations or minor engineering failures (Zavala-Araiza et al., 2017), while others stem from large unexpected leaks, (e.g. (Conley et al., 2016; Ryerson et al., 2012)). In some
O&G fields, large amounts of non-recoverable $CH_4$ can be flared or vented due to a number of factors. Thus, the composition of O&G emissions can be influenced by several variables, including the targeted hydrocarbon product (oil or gas), extraction techniques and gas capture infrastructure. O&G installations co-emit volatile organic compounds (VOCs) such as alkanes, alkenes and aromatics in addition to $CH_4$. Some of these VOCs are toxic and can have direct health impacts or, together with $NO_x$ can produce ozone, having an impact on the regional air quality (Edwards et al., 2013). VOC and $\delta^{13}C_{CH4}$ measurements
can be utilised to fingerprint the main processes or likely location responsible for associated $CH_4$ emissions (Cardoso-Saldaña et al., 2019; Lee et al., 2018; Yacovitch et al., 2014a). A recent study has also demonstrated the cost-effectiveness of airborne measurements for leak detection and repair at O&G facilities relative to traditional ground-based methods (Schwietzke et al., 2019).

There is thus a need to develop reliable methodologies to locate emissions, pinpoint sources and design effective mitigation
action in sufficient detail to allow quantification of emissions and validate against publicly reported inventory emissions. To date, a number of approaches have been trialled. Airborne measurements of both individual and clusters of facilities, along with production data, have been used to scale up to an inventory of $CH_4$ emissions for the U.S. Gulf of Mexico (Gorchov Negron et al., 2020). Ship based measurements of $CH_4$ and associated source tracers have been made in both the Gulf of Mexico (Yacovitch et al., 2020) and in the North Sea (Riddick et al., 2019). The latter reported fluxes of $CH_4$ from offshore
O&G installations in UK waters that were derived from observations made from small boats at ~2 m above sea level. This approach has advantages in terms of cost, but the authors recognised a number of key uncertainties in their approach associated with assumptions around boundary layer conditions and a lack of 3D information (i.e. Gaussian plume modelling and assumptions of constant wind speed). Measurements from aircraft can provide this 3D spatial information enabling better characterisation of both plume morphology and boundary layer dynamics.

Here we report a project that was designed around the use of a small-aircraft with flexible instrument payload suitable for agile deployment. Key objectives were i) to identify and quantify emissions of $CH_4$ from a suite of offshore gas fields within a limited geographical area, and ii) to develop methodologies that can be applied to gas fields elsewhere to assess emissions at local scales. The project was part of the United Nations Climate and Clean Air Coalition (UN CCAC) objective to characterize





global $CH_4$ emissions from oil and gas infrastructure. Targeted observations of atmospheric $CH_4$ and $C_2H_6$, plus sampling for VOC and $\delta^{13}C_{CH4}$ analysis were made from a Twin Otter aircraft operated by the British Antarctic Survey. Two campaigns were conducted, one in April 2018 and one in April/May 2019, with a total of 10 flights (~45 hours) over the two campaigns. The specific aims of the surveys were:

1. $CH_4$ surveying of facilities with a range of expected (from inventories) $CH_4$ emissions.

2. Resolution of types of emission from installations (such as flaring, venting, combustion and leaks) using carbon-isotopic fingerprinting and analysis of co-emitted species (including VOCs).

3. Estimation of total $CH_4$ emissions for the target region.

4. Improvement of flux estimation (and sampling) methodologies for rapid quantification of major gas emissions.

Here, we provide an overview of measurement platform configuration and sampling strategy during these campaigns, including instrument comparisons for hydrocarbon plume detection, spot sampling strategies for VOCs and $\delta^{13}C_{CH4}$, and flight planning to cope with complex boundary layer meteorology to allow estimation of emission fluxes. Analysis methods to determine diagnostic hydrocarbon plume characteristics such as $C_2H_6$-$CH_4$ ratios and $\delta^{13}C$ source attribution are also discussed. A sister publication will present the estimated facility-level emissions in detail and discuss the results in a regional context.

## 2. Experimental

A DHC6 Twin Otter research aircraft, operated by the British Antarctic Survey, was equipped with instrumentation to measure atmospheric boundary layer parameters, including the boundary layer structure and stability, as well as a number of targeted chemical parameters. These included $CH_4$, $CO_2$, $H_2O$, $C_2H_6$ as well as whole air sampling for subsequent analysis of $\delta^{13}C_{CH4}$ and a suite of VOC's. Here we describe the aircraft capability, aircraft fit, and the instruments deployed.

### 2.1. Aircraft capability

The maximum range of the Twin Otter aircraft during the flight campaigns was approximately 1000 km. Although the aircraft is capable of flying up to 5000 m altitude, most of the flying was limited to below 2000 m; in regions with no minimum altitude limit, the aircraft could be flown at the practical limit of 15 m (~50 ft) above sea level. The instrument fit included use of a turbulence boom, which limited the speed to a maximum of 140 kts (~70 $ms^{-1}$); throughout the campaigns, the target aircraft speed for surveying was 60 $ms^{-1}$. The aircraft was limited to a minimum safe separation distance of 200 m from any O&G production platforms.

The total weight of the aircraft on take-off is limited to 14,000 lbs (6,350 kg). Allowing for fuel and crew, this left 2,086 kg for the instrumentation. The total power available on the aircraft is 150 A at 28 V and inverters were used to provide 220 V to those instruments that required it.



Altitude and air speed were determined by static and dynamic pressure from the aircraft static ports and heated Pitot tube, logged using Honeywell HPA sensors at 5 Hz. A radar altimeter recorded the flight height at around 10 Hz. An OXTS Inertial

measurement system coupled to a Trimble R7 GPS was used to determine the aircraft position and altitude. This system gives all three components of aircraft position, altitude and velocity at a rate of 50 Hz.

The chemistry inlets on the Twin Otter are similar to those fitted to the FAAM BAE 146 large atmospheric research aircraft (e.g. O'Shea et al., (2013)) and were fitted with the inlet facing to the rear (Fig. A1). A single line (¼" Synflex tubing) was taken from the inlet to a high capacity pump with the instruments branching from this line. This approach was taken to minimise

the delay between air entering the inlet and reaching the instruments, which, with this configuration, was ~5 seconds.

The aircraft was fitted out during the week before each of the two flight campaigns allowing significant changes to be made between 2018 and 2019 based on instrument performance and data from 2018 (Fig. 1).

### 2.2. Boundary layer physics instrumentation

A fast response temperature sensor and a nine hole NOAA BAT 'Best Air Turbulence' probe (Garman et al., 2006) was mounted

on a boom on the front of the aircraft (see photo, Fig. A2). This instrumental set-up was chosen to reduce flow distortion effects by the aircraft. These fast response measurements of wind and temperature fluctuations were made with a frequency of 50 Hz. Garman et al. (2006) investigated the uncertainty of the wind measurements by testing a BAT probe in a wind tunnel. They assessed that the precision of the vertical wind measurements due to instrument noise was approximately $\pm 0.03$ ms$^{-1}$. Garman (2008) showed that an additional uncertainty in the wind data occurs when a constant up-wash correction value is

used, as proposed by the model of Crawford (1996). We use the Crawford model which increases the uncertainty in the vertical wind component, w, to approximately $\pm 0.05$ ms$^{-1}$. We assume for the two horizontal wind components, u and v, similar high uncertainties due to aircraft movement. A detailed description of the Twin Otter turbulence instrumentation and associated data processing can be found in Weiss et al. (2011).

Ambient air temperature was observed with Goodrich Rosemount Probes, mounted on the nose of the aircraft. A non-de-iced

model 102E4AL and a de-iced model 102AU1AG logged the temperature at 0.7 Hz. Atmospheric humidity was measured with a Buck 1011C cooled mirror hygrometer. The 1011C Aircraft Hygrometer is a chilled mirror optical dew point system. The manufacturer stated the reading accuracy of $\pm 0.1$ °C in a temperature range of -40 to +50 °C. Chamber pressure and mirror temperature were recorded at 1 Hz.

### 2.3. *In situ* atmospheric chemistry instrumentation

A Los Gatos Research (LGR) Ultraportable Greenhouse Gas Analyser (uGGA) was installed to measure $CH_4$, $CO_2$ and $H_2O$. Expected manufacturer precision for the $CH_4$ measurement was < 2 ppb averaged over 5 s and < 0.6 ppb over 100 s. The response time of the LGR uGGA itself (i.e. the flush time through the measurement cell) was 10 s. To achieve higher temporal frequency data, a fast Picarro G2301-f was installed that provided measurements of $CH_4$, $CO_2$ and $H_2O$ at ~10 Hz, with 1-σ precision at 10 Hz of < 3 ppb for $CH_4$. A third greenhouse gas analyser, a LGR Ultraportable $CH_4/C_2H_6$ Analyser (uMEA) was



used to measure $CH_4$ and $C_2H_6$. The 1-σ precision of the uMEA, as stated by the manufacturer, was < 2 ppb for $CH_4$ and < 30 ppb for $C_2H_6$ at 1 s, however in-house laboratory measurements suggest $C_2H_6$ 1-σ precision at 1 s is ~17 ppb for this unit. During the 2019 airborne campaign, atmospheric $C_2H_6$ was monitored by a Tuneable Infrared Laser Direct Absorption Spectrometer (TILDAS, Aerodyne Research, Inc) (Yacovitch et al., 2014b). This instrument utilises a continuous wave laser operating in the mid-infrared region (at λ = 3.3 μm). Further description of the TILDAS instrument set-up and performance is

available in the Appendices.

### 2.4. Calibration of *in situ* instrumentation

### 2.4.1. $CH_4$ and $CO_2$ calibration

In-situ $CH_4$ and $CO_2$ instruments were calibrated in-flight using a manually operated calibration deck, shown in schematic form in Fig. 2. The calibration gases consisted of a suite of WMO referenced standards with a "High", "Low" and "Target"

designation. The "High" $CH_4$ concentration was ~2600 ppb, the "Low" ~1850 ppb and the "Target" ~2000 ppb concentration. $CO_2$ concentrations were "High" ~468.5 ppm, the "Low" ~413.9 ppm and the "Target" ~423.6 ppm. The absolute values of the cylinders varied between years as they were re-filled and re-certified to the NOAA WMO-$CH_4$-X2004A and WMO-$CO_2$-X2007 scales. The calibration deck is designed so that upon the calibration valve opening, the calibration gas flow rate is sufficient to overflow the inlet. A similar approach to in-flight calibration is also applied on the NOAA WP-3D aircraft

(Warneke et al., 2016). Full details of the calibration procedure is recorded in the Appendices. $CH_4$ uncertainty (1σ) at 1 Hz is calculated as 1.24 ppb for the Picarro G2301-f and 1.77 ppb for the uGGA, giving performance comparable with similar instrumentation on the FAAM aircraft (O'Shea et al., 2014). The excellent agreement between measured and expected values of $CH_4$ for the target cylinder (for the Picarro and uGGA) gives us confidence in being able to operate to high levels of accuracy with a very limited period of instrument fitting and testing. $CO_2$ uncertainty (1σ) at 1 Hz is calculated as 0.20 ppm for the

Picarro G2301-f and 0.35 ppm for the uGGA. More details on the calibration and associated uncertainties are shown in the Appendices.

### 2.4.2. $C_2H_6$ calibration

The calibration cylinders installed on the Twin Otter during both campaigns did not contain a sufficient range of $C_2H_6$ concentrations to enable calibrations to be performed. This represents a limitation on the accuracy and traceability of the $C_2H_6$

measurements during these campaigns and will be addressed for future studies using the BAS Twin Otter. The uMEA was calibrated in the laboratory post campaign for the 2018 campaign, and pre-and post-campaign in the laboratory for the 2019 season. Corrections for $C_2H_6$ and $CH_4$ measurement drift as a function of cavity temperature were determined experimentally by analysing two calibration cylinders alternately over the course of several hours as the cavity temperature increased. These corrections were then applied to the uMEA $C_2H_6$ and $CH_4$ measurements obtained from both 2018 and 2019 flight campaigns.





The TILDAS (deployed in 2019) was corrected for water vapour using the TDLWintel software (Nelson et al., 2004) to account

for changes in humidity during the flight (as discussed in Pitt et al., (2016)). The instrument has a quoted precision of 50 ppt

for an averaging time of 10 s. The raw measured data were calibrated pre- and post-flight using two cylinders of known

concentration, whose mole fractions spanned the measurement range observed during flights for $C_2H_6$. By assuming a linear

relationship, the calibrated mole fraction corresponding to each measured TILDAS mole fraction was given by interpolating

the scale between the pre- and post-flight calibration reference points.

Previous studies have reported the sensitivity of TILDAS systems to aircraft cabin pressure (e.g. Pitt et al., (2016) and

Gvakharia et al., (2018)). This sensitivity means that the $C_2H_6$ mole fractions measured during the flight contain a systematic

altitude-dependent bias. However, as cabin pressure only affects the spectroscopic baseline, the zero-offset of the

measurements is affected, but not the instrument gain factor. Therefore, as long as each plume measurement is referenced to a

measured background at the same altitude, this cabin pressure sensitivity does not significantly impact the calculated $C_2H_6$

mole fraction enhancements. As stated above, future deployments will mitigate this issue by employing in-flight calibration

cylinders that are certified for $C_2H_6$. The potential to use a fast, frequent calibration for baseline correction as described by

Gvakharia et al., (2018) will also be investigated, although this has payload implications as it requires an extra calibration

cylinder.

**2.5. Spot Sampling**

On a rapid aircraft fit, manually triggered spot sampling provides a cost-effective and simple method for analysis which cannot

be performed mid-flight or require specialist laboratory facilities to gain useful levels of precision. Two discrete air-sampling

systems were used during these flights to enable post-flight analysis for VOCs and $\delta^{13}C_{CH4}$.

**2.5.1. Son of Whole Air Sampler (SWAS)**

The Son of Whole Air Sampler (SWAS) is a new, updated version of the parent WAS system fitted to the FAAM BAE 146

large atmospheric research aircraft (e.g. as used in O'Shea et al., (2014)), which it is designed to supersede. The system

comprises a multitude of inert Silonite-coated (Entech) stainless steel canisters, grouped together modularly in cases with up

to 16 canisters per case. On-board the Twin Otter, 2 cases can be fitted allowing up to 32 canisters to be carried per flight. The

theory of operation is to capture discrete air samples from outside of the aircraft and compress the sample either into 1.4 L or

2 L canisters at low pressure (40 psi) via pneumatically-actuated bellows valves (PBV, Swagelok BNVS4-C). Full details of

the operation of SWAS are included in the Appendices. For the 2019 campaign, SWAS was updated with the addition of 2 L

flow-through canisters making narrow plumes easier to capture due to reduced sample line lag and fill times.

SWAS canisters sampling was manually triggered during the flights according to in-situ observations made by fast response

instrumentation such as $CO_2$, $C_2H_6$ and $CH_4$, with the aim of capturing specific oil and gas plumes. The samples were analysed

at the University of York for VOCs post-flight using a dual-channel gas chromatograph with flame ionisation detectors

(Hopkins et al., 2003). Firstly, 500 mL aliquots of air are withdrawn from the sample canister and dried using a condensation



finger held at -30 °C then pre-concentrated onto a multi-bed carbon adsorbent trap consisting of Carboxen 1000 and Carbotrap B (Supelco), and transferred to the GC columns ($Al_2O_3$, $NaSO_4$ deactivated and open tubular PLOT) in a stream of helium. Chromatogram peak identification was made by reference to a calibration gas standard (NPL30, D600145 - 2018) containing

known amounts of 30 VOCs ranging from C2-C9. Compounds of interest include $C_2H_6$, propane, butanes, pentanes, benzene and toluene; a full list is shown in Table A1.

### 2.5.2. Flexfoil Bag Sampling

Spot sampling for $\delta^{13}C_{CH4}$ by collecting whole air samples into Flexfoil bags (SKC Ltd) has been in use on both the FAAM BAE 146 research aircraft (e.g. (Fisher et al., 2017)) and during ground based mobile studies (e.g. (Lowry et al., 2020) and

provides a relatively cost-effective and rapid methodology for sample collection. The method does have some limitations, however, as the Flexfoil sample bags are only stable for a number of compounds (including $CH_4$) and is not a true whole air sample. Samples captured in both Flexfoil bags and SWAS were measured at Royal Holloway using CF-IRMS (continuous flow isotope ratio mass spectrometry) (Fisher et al., 2006) and each measurement has a $\delta^{13}C_{CH4}$ uncertainty of ~0.05 ‰. Each sample is also measured for $CH_4$ mole fraction using cavity ring-down spectroscopy to allow direct comparison to in-flight

data (Fig A3). Alternative, continuous in-flight $\delta^{13}C_{CH4}$ instrumentation currently cannot replicate the precision of laboratory sampling, and the few seconds of enhanced $CH_4$ that would be encountered during flight is not sufficient for averaging of continuous $\delta^{13}C_{CH4}$ data to gain a meaningful source $\delta^{13}C_{CH4}$ signature (e.g. (Rella et al., 2015)).

### 3 Overall approach to flight planning

The majority of flights were conducted during good operating conditions i.e. - daytime, no precipitation, clear or broken cloud,

winds < 10ms⁻¹ and visibility to allow flying at minimum safe altitude around the task area. Two approaches were trialled to assess $CH_4$ emissions from offshore gas installations: (1) regional survey, and (2) specific plume sampling. The flight modes are demonstrated in Fig. 3, with the dark grey pattern showing a flight plan for regional measurements and the orange and white patterns demonstrating specific plume sampling flight patterns. Flight plans to sample specific installations were designed to capture a full range of expected emissions using the UK National Atmospheric Emissions Inventory (NAEI) as a

guide.

Regional survey intentions were two-fold: firstly, to offer an identification process for emitters of interest that could specifically be targeted for plume sampling modes, and secondly, to build a picture of aggregate bulk emissions for multiple upwind platforms. This method has been successfully employed during a Gulf of Mexico airborne study (Gorchov Negron et

al., 2020). However, in this instance, regional surveys were poor for identifying plumes (being too far downwind of platforms or not intercepting thin filament layers containing $CH_4$ enhancements) and attempts to aggregate bulk emissions were hindered by the often encountered complex boundary layer structure over the area, which controlled dispersion of $CH_4$ emissions from



rigs. From the regional flight data derived in 2018, and considering the work in other offshore studies in this area (e.g. Cain et al., (2017)), the regional flight mode was determined to be of limited scientific value in the context of this project and this flight pattern was not used during the 2019 campaign.

Plume sampling flights were conducted in both 2018 and 2019. These flights involved the use of a box pattern to create both upwind and downwind transects either side of the infrastructure of interest. Upwind transects provided an understanding of other methanogenic sources (such as other installations, ships or long range transport of air masses from on-shore sources) that could interfere with observed $CH_4$ plumes downwind, and were conducted to be confident that plumes were solely originating from the targeted infrastructure. Vertically stacked downwind transects were conducted to better capture the vertical extent of the plume in a 2-dimensional Lagrangian plane for $CH_4$ flux quantification using mass balance analysis (e.g. O'Shea et al., (2014)). The vertically stacked transects in profile, as planned from the 2019 field deployment, are demonstrated in Fig. 3. The separation between vertically stacked transects was usually 200 ft (60 m) with a minimum absolute height of 150 ft (45 m) above sea surface up to approximately 850 ft (260 m) to capture the entire extent of a downwind plume. Plume dispersion was dependent on meteorology and emission type (venting, fugitive, or combustive emissions) and as such, maximal plume heights varied between individual infrastructure. Upwind transects were flown at a median height between the minimum and maximum stacked runs.

**4. Assessing and addressing issues encountered during flights**

A number of issues were encountered during the flights that influenced the measurements made. An initial presentation of these issues is given here, with recommendations for improvements given in Section 6 below.

**4.1. Complex marine boundary layers**

Boundary layer structure proved to be a decisive influence on observed $CH_4$ mixing ratios. Figure 4 shows the measured profiles of $CH_4$ (left hand panel) and potential temperature (right hand panel) during an off-shore flight in April 2018 along with the corresponding synoptic chart. Potential temperature was calculated as described in Stull (1988). The potential temperature profile demonstrates that the boundary layer structure on this day (and many other days) was partly stable stratified, showing mostly an increase in potential temperature with height, and the boundary layer showed complex layering. The prevailing meteorological situation at that time, illustrated by the synoptic chart in Figure 4, was of a persistent anticyclonic ridge, stretching from the south-west over the British Isles and Western Europe, with associated low wind speeds and poorly defined air flow over the southern North Sea sector. The observed layering was partly also caused by residual boundary layers from previous days and nights which had not dispersed. The structure of the boundary layer in Fig. 4 clearly had an important influence on the vertical profile of $CH_4$, which varied and shows a complex profile with height. Due to the complexity of the



boundary layer structure, it was concluded that it would be inappropriate to use a particle dispersion model such as NAME (Jones et al., 2007) to derive a bulk regional emission estimate.

The impact of the residual layers of $CH_4$ enhancement make in-flight decision very challenging for two main reasons: 1. The difficulty of determining which enhancements are from installations and require further investigation, especially if flying at some distance downwind from a potential source or on a regional survey pattern. 2. Emissions being actively released can become trapped in vertically thin filaments, which can be easily missed when flying stacked legs, depending on flight altitude. In contrast, on days with a well-mixed boundary layer the $CH_4$ profile stays relatively constant with height, and shows increase

only in the surface layer near $CH_4$ sources. Fig. 5 shows an example of composite $CH_4$ and potential temperature profiles flown within an hour of each other, in a well-mixed boundary layer during a flight in May 2019; the synoptic situation on that day was consistent with a slow-moving cyclonic south-easterly air flow. It can clearly be seen how the potential temperature and $CH_4$ profiles stay almost constant with height above 200 m, and only show structure near $CH_4$ sources. The potential temperature profile indicates neutral stratification of the boundary layer.

**4.2. Instrument response times**

The role of the continuous in-flight measurements is to provide the backbone of the dataset and ensure that, at a bare minimum, the flights are able to identify areas of $CH_4$ enhancement and inform on the likely sources of the $CH_4$ enhancement, hence the decision to run redundancy measurements of $CH_4$ utilising an LGR uGGA. Figure 6 shows a typical suite of instrument response to a $CH_4$ plume and it is clear that the cell turnover time of the uGGA is not sufficient to capture the fine detail of

the plume. Whilst the uGGA and uMEA are capable of determining whole infrastructure mass balance and average infrastructure ethane-methane ratios, the refined understanding of the true plume is lost in these slower response instruments. This is important as the combined Picarro G2301-f and TILDAS data can detect several sources from the same installation (Fig. 6), because of their rapid measurement cell turnover. This information can be used to infer either cold venting ($CH_4$ & $C_2H_6$) or combustion from flares or generators ($CO_2$, $CH_4$ and $C_2H_6$) which could then be used to determine $CH_4$ emission

factors from identified flares (Gvakharia et al., 2017).

There are a number of other implications that arise from slow measurement response. For example, in-flight spot sampling requires guidance from fast response instruments that can indicate the optimum timing to collect samples that span the plume, and thereby capture the representative chemical nature of the plume. Further, in-flight calibrations must be matched to the

slowest response instrument to ensure stabilisation of the measurement of calibration gases across all instruments. Use of slower-response instruments thus induces additional, unwanted loss of measurement time and excessive use of calibration gases and the benefits of instrument redundancy should be carefully considered.



### 4.3. Spot sampling improvements between 2018 and 2019 campaigns

In-flight spot sample collection was carried out during both the 2018 and 2019 campaigns. Such sampling is challenging, and requires fast response instruments to be viewable to the operator to give the best chance of collecting samples at appropriate times that span the plume. For 2019, a number of simple adaptations were introduced that significantly increased the success of capturing plumes (Fig. A3). The improvements included modified flight planning, with an increased number of passes through discovered plumes. This approach resulted in increased fuel consumption per plume, but contributed to the higher success rate of plume capture. The comprehensive update to the SWAS system, which included continuous sample through-flow allowed more precise spot sampling to be achieved.

### 5. Creation of data products

### 5.1. Methane fluxes

A methane flux can be calculated from the $CH_4$ mixing ratio data using mass balance techniques (e.g. (O'Shea et al., 2014; Pitt et al., 2019) in which a vertical 2D plane is defined at a fixed distance downwind of the infrastructure of interest, and sampling is conducted across the stacked transects at this distance if a plume is identified in the downwind plane. Fluxes were derived using Eq. (1):

$$Flux = \left( X_{plume} - X_{background} \right) \times n_{air} \times V \times \Delta x \tag{1}$$

where *Flux* is the bulk net flux passing through the plane per metre altitude (mol m$^{-1}$ s$^{-1}$), $n_{air}$ is the molar density of air (mol m$^{-3}$), $X_{plume}$ is the average $CH_4$ mole fraction measured within the plume, and $X_{background}$ is the $CH_4$ mole fraction of the background. *V* is the wind component perpendicular to the flight track and $\Delta x$ is the plume width perpendicular to upwind-downwind.

The $CH_4$ and $CO_2$ measurements from the 10 Hz response instruments were used to provide the highest accuracy in (1) lateral plume width and (2) number of unique plumes identified from each individual platform. Slower response instruments would allow for flux calculations but would not be able to identify individual plumes from the same platform. This could be useful to distinguish, for example, multiple plumes from different emission processes that are spatially distinct within the same platform (e.g., a fugitive source vs. a flare). A background mixing ratio was selected to best represent the conditions observed during the flight at the specific time of survey. An average of 30 s of data either side of the plume on each run was used, if this was deemed appropriate with a clean upwind sampling leg. When the upwind sampling was contaminated, more caution should be taken when selecting an appropriate background so that the background value is not distorted by extraneous far-field sources.



For this analysis, a flux across each individual vertical run downwind of a plume was calculated before scaling in the vertical component. The flux was then integrated across potential minimum and maximum plume depths. Figure 7 (upper panel) represents a reduced vertical resolution of the plume where transects at intermediate altitudes through the plume were not conducted. In this case, the minimal plume depth is the narrow span captured by observation in the 45.9 – 51.9 m altitude window. The maximal plume depth is taken as the height difference between the maximal and minimal altitude transects that

do not demonstrate $CH_4$ enhancements so are outside of the plume; this value has to be used as the maximum due to incomplete sampling of the void area seen in the upper panel of Fig. 7. In cases where the base and top of the plume were not sampled (e.g. during 2018 sampling), the lower limit was selected as the sea surface and the upper limit of the plume was selected as the atmospheric marine boundary layer. The greatest uncertainty in bulk flux arises when the vertical resolution is not fully captured. Between the 2018 and 2019 portions of the campaign, the flux uncertainty related to plume depth was reduced by a

factor of 10 (as seen in Table 1) by completing a rigorous set of stacked transects at multiple heights throughout the plume.

### 5.2. Ethane-Methane ratios (C2:C1) as a source tracer

It has already been well established that continuous $C_2H_6$ measurements can be an excellent diagnostic tool for ascribing enhancements of co-located $CH_4$ and $C_2H_6$ to natural gas emissions in both urban areas (e.g. (Plant et al., 2019)), semi-rural areas (e.g. Lowry et al., 2020)) and during large scale evaluations of oil and gas fields from aerial studies in the USA (e.g.

(Peischl et al., 2018)), Canada (Johnson et al., 2017), and the Netherlands (Yacovitch et al., 2018). During this work, two methods were used to establish $C_2H_6$-$CH_4$ ratios (hereafter, described as C2:C1). In 2018 a Los Gatos ultraportable $CH_4/C_2H_6$ analyser (uMEA) was used. The benefits of such instrumentation are in its simplicity of operation and that few considerations are required for corrections or variable lags as both species are measured at the same rate and within the same optical cavity. C2:C1 can therefore be readily determined as the gradient of a linear regression between the $C_2H_6$ and $CH_4$ measurements.

However, the low sensitivity to $C_2H_6$ (standard deviation of ~10 ppb in $C_2H_6$ over 10 s of background flying) only allowed emissions from two platforms to be characterised for C2:C1 ratios during the whole of the 2018 flying, and none during 2019.

In 2019 the addition of the TILDAS 1 Hz $C_2H_6$ instrument allowed for better precision of $C_2H_6$ (< 1 ppb) with a faster flush time in the measurement cell. The $C_2H_6$ data is time matched with the 1 Hz Picarro $CH_4$ data set to allow C2:C1 derivation.

As the instruments do not have the exact same flow rate and different cell residence times, the C2:C1 ratios were determined using the integral of each $CH_4$ and $C_2H_6$ enhancement using Gaussian peak fitting. A comparison between the 2018 flight, 2019 flight and published data derived from the same geographical area, is shown in Table 2. Although both instruments have been operated for this work without in-flight calibration or engineering solutions to address cabin pressure sensitivity issues (Gvakharia et al., 2018) due to weight and time constraints, the agreement between years and with published expected values

is highly reassuring. The added value in high precision C2:C1 demonstrates that $C_2H_6$ is not just a tracer for matching emissions to natural gas; it can give information as to proportions of emissions from mixed sources (as previously used in Peischl et al.,





(2018)) or can be used to identify a likely emission point in a process chain depending upon enrichment or depletion of $C_2H_6$ relative to $CH_4$. The inclusion of a continuous instrument with a sub-ppb level of detection for $C_2H_6$ is considered vital for future work with thermogenic sources of $CH_4$ to allow more precise source attribution of emissions where no spot sampling has occurred.

### 5.3. $\delta^{13}C_{CH4}$ for $CH_4$ source attribution

The principal method of $\delta^{13}C_{CH4}$ source characterisation utilises the principles outlined in Keeling (1961) and Pataki et al., (2003), and has been well utilised since to create $\delta^{13}C_{CH4}$ databases for a plethora of known $CH_4$ sources (e.g. (Sherwood et al., 2017)). In order for a Keeling plot to give useful results to determine a $\delta^{13}C_{CH4}$ source signature of a $CH_4$ emission, the emission must have been successfully captured multiple times and ideally with a range of $CH_4$ mixing ratios (which could be achieved by passes at different distances or heights downwind of a point source). This sampling process takes time (especially on an aircraft), where the emission plume is only intercepted once per transect and time in the plume is limited so that only one spot sample can be taken whilst "in-plume". Beyond the time limitations, sampling of a range of $CH_4$ mixing ratios from emissions and appropriate background samples is not straightforward. Background sampling must capture the air into which emissions are released, but during flights the meteorological conditions often resulted in significant variation of $CH_4$ mixing ratio and $\delta^{13}C_{CH4}$ with altitude, in addition to horizontal variations. Where repeat transects were conducted at different altitudes, this made selection of appropriate background samples for Keeling plots challenging, since the background $CH_4$ mixing ratio and $\delta^{13}C$ varied over the different altitudes. This becomes particularly detrimental to Keeling plot validity where the range in sampled emission mixing ratios is small, since uncertainty in the background samples then becomes more important.

In Fig. 8, a sensitivity analysis is presented from one of the flights investigating the effect of reducing the number of samples on the uncertainty in the $\delta^{13}C_{CH4}$ source signature determined for a plume. In this case nine samples were collected, but over eight downwind transects and one upwind transect of a cluster of installations, which is not feasible to repeat for sampling large numbers of installations. As shown in Fig. 8, the uncertainty in the $\delta^{13}C_{CH4}$ source signatures increases only slightly with reduction in number of sampling points, with the exception of one n = 3 run where the source signature is poorly defined. A minimum of three data points can be therefore sufficient for classifying a source of $CH_4$ emissions (such as thermogenic, microbial or pyrogenic sources), providing that the background and point samples are captured with a large enough range of $CH_4$ concentration, and providing that there is no mixing of sources. This will typically require collection of more than three samples, given some may miss the targeted plumes or potentially be lost during storage/processing as aforementioned. Although a two-point Keeling plot is technically possible, it is impossible to gauge the quality of the regression to be sure that only a single source has been captured.



## 6.   Conclusions

Given the restrictions and time constraints on the science flights, important lessons for offshore oil and gas airborne measurement campaigns have been learned for rapid instrument re-fitting and agile deployment of a small aircraft for future

campaigns. A key finding from this study is that offshore meteorological conditions define the ability of the flights to produce valuable data, and suitable meteorology with a well-mixed (neutral) boundary layer is critical to deriving a regional emission estimate through regional modelling. Flying in conditions with multiple residual boundary layers makes interpretation difficult and pin-pointing emissions especially challenging as emission plumes can easily be missed when they are trapped in thin filaments, dramatically increasing the uncertainties of measurement-based emission flux calculations. Although not possible

for this work given aircraft scheduling, it is recommended that offshore observations are scheduled with a long window of opportunity to ensure optimal flying conditions. Predictions of the likelihood of a residual boundary layer over a coastal area could be achieved through high spatial resolution forecast models such as the UK Met Office forecast model (Milan et al., 2020). Information on the temperature structure over the previous few days using all the assimilated information, such as tephigrams and synoptic charts, would help determine the likelihood of residual boundary layers versus a simpler stratified,

well mixed layer. For methods using alternative platforms such as ships or drones, co-incidental measurements of vertical profiles must be made to capture the true nature of the emission plume in the current meteorology.

Unlike some of the larger aircraft, payload restrictions and power limitations demand challenging decisions for instrument selection. We recommend deploying at least one instrument measuring $CH_4$ (and $CO_2$) at 10 Hz, allowing several plumes emitted from a single installation to be resolved (Fig. 6). Priority should next be given to a $C_2H_6$ measurement capable of sub

ppb limit of detection at 1 Hz (or higher) in order to give certainty to the source of the $CH_4$ emission. Using C2:C1 appears to be the simplest method for source attribution, and is robust for distinguishing natural gas emissions, where the gas has an $C_2H_6$ component (Lowry et al., 2020; Plant et al., 2019). Spot sampling is challenging, payload heavy and time consuming as several passes are needed to collect enough samples (especially for $\delta^{13}C_{CH4}$ source attribution). However, results can be very

informative such as the ability to distinguish a gas leak geological reservoir from depth or near surface (Lee et al., 2018). The improvements to SWAS, allowing for continuous through flow, has increased the success rate of peak sampling, but still relies on accurate user triggering.

For mass balance flux calculations, an instantaneous plume and the surrounding background variation in the species of interest,

alongside local meteorology, must be fully resolved during the observation stage. This includes instruments with appropriate response times to fully capture the plume and identify any internal structure that may suggest a mixed source. An upwind leg must be conducted to ensure the plume and background is not contaminated by extraneous far-field sources and the plume must be significantly distinct from this background for meaningful flux calculations. The plume must be laterally and vertically resolved in the 2D plane as much as possible at a fixed distance downwind of the source. Straight and level runs must extend





either side of the plume and the vertical resolution must include multiple stacked transects with an identification of the top and
bottom of the plume (where feasible) to reduce uncertainty in the plume bulk net flux. Full understanding of the meteorology
during data collection with meteorological measurement instrumentation and an entire profile to determine the marine
boundary layer characteristics from the top to the surface, including determination of inversion heights, must be conducted
during the flight day when appropriate radiosonde soundings are not available. The observed impact of complex boundary

layer dynamics on plume dispersion also highlights an important limitation of ship-based plume measurements, which are
unable to resolve the vertical structure of the plume and therefore rely on the assumption of idealised models of plume
dispersion.



**7. Figures and Tables**







Figure 1. Instrument schematics for the Twin Otter aircraft as deployed in 2018 and 2019, detailing changes in layout and instrumentation between the two campaigns. The top panel is the 2018 fit and lower panel 2019 fit.




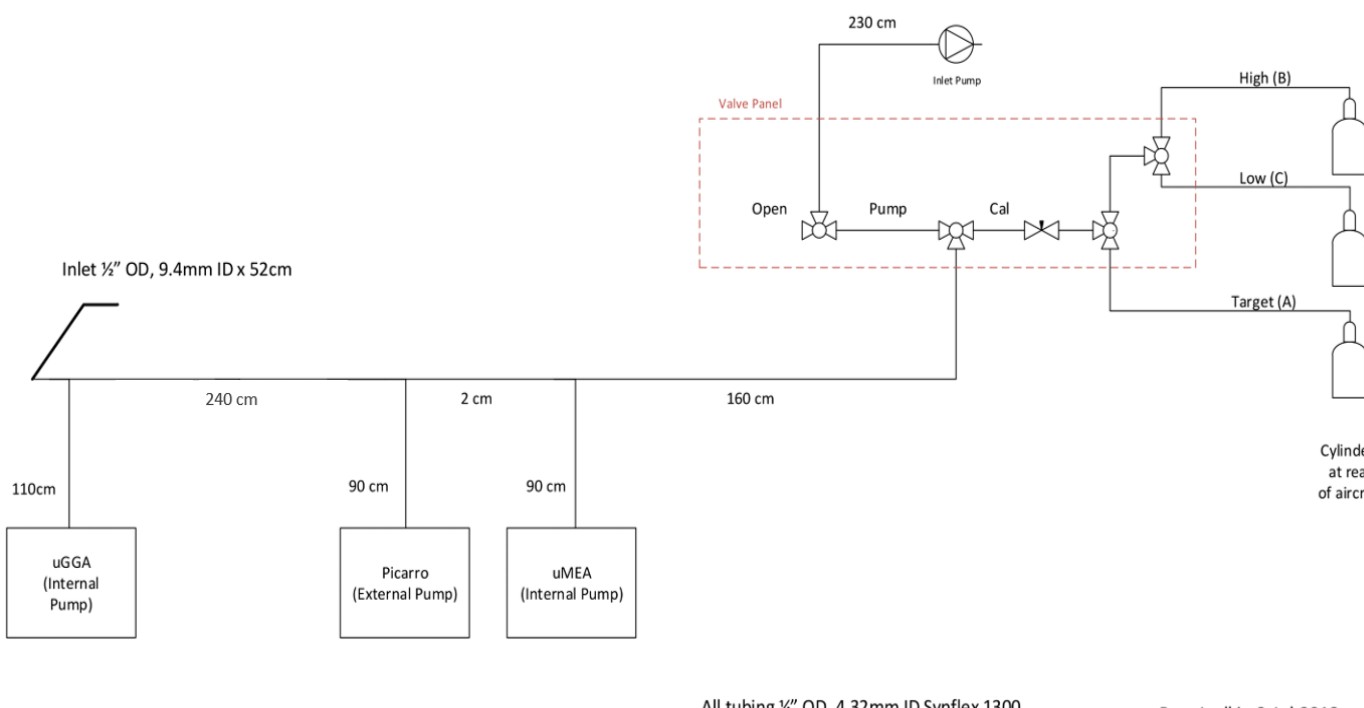

**Figure 2. Layout of plumbing of the calibration system (and inlet system) for 2018 campaign.**





**Figure 3: [top panel] flight patterns showing the regional and plume capture styles of flight deployed between 2018 and 2019, alongside infrastructure of interest (such as drilling rigs, gas distribution platforms or production platforms). [bottom panel] a 2019 plume sampling survey showing idealised stacked transects in the 2D plane downwind of infrastructure of interest.**

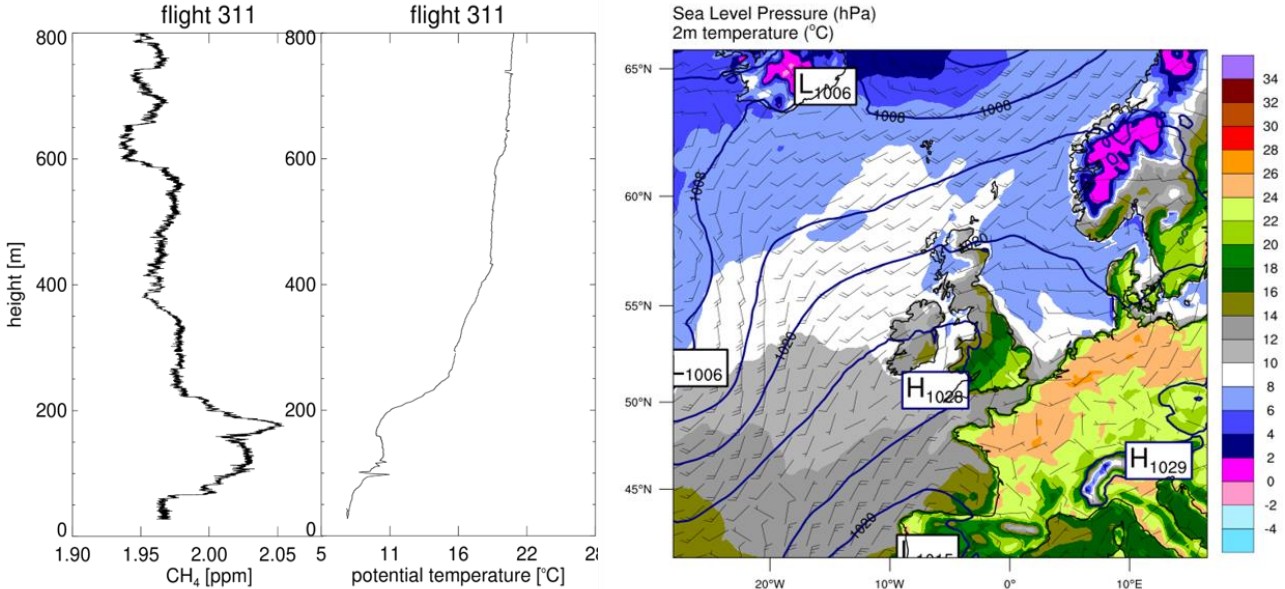

**Figure 4. Example of CH₄ and potential temperature profiles showing the large amount of structure arising from residual boundary layers. The increase of the potential temperature with height shows stable stratification of the boundary layer. The synoptic chart over the eastern North Atlantic and north West Europe shows contoured sea level pressure (hPa), 2m temperature (°C, right-hand side colour scale) and wind for 20/04/2018 12:00 UT, and reveals relatively low wind speeds and poorly defined air flow over the southern North Sea sector, allowing the build-up of residual boundary layers. Synoptic chart image produced by the UK National Centre for Atmospheric Science (NCAS) using Weather Research and Forecasting model WFR-ARW version 3.7.1, with a 20 km grid spacing, 51 vertical levels initialised using the NOAA Global Forecast System. NCAS Weather Research Catalogue (sci.ncas.ac.uk/nwr/pages/home)**



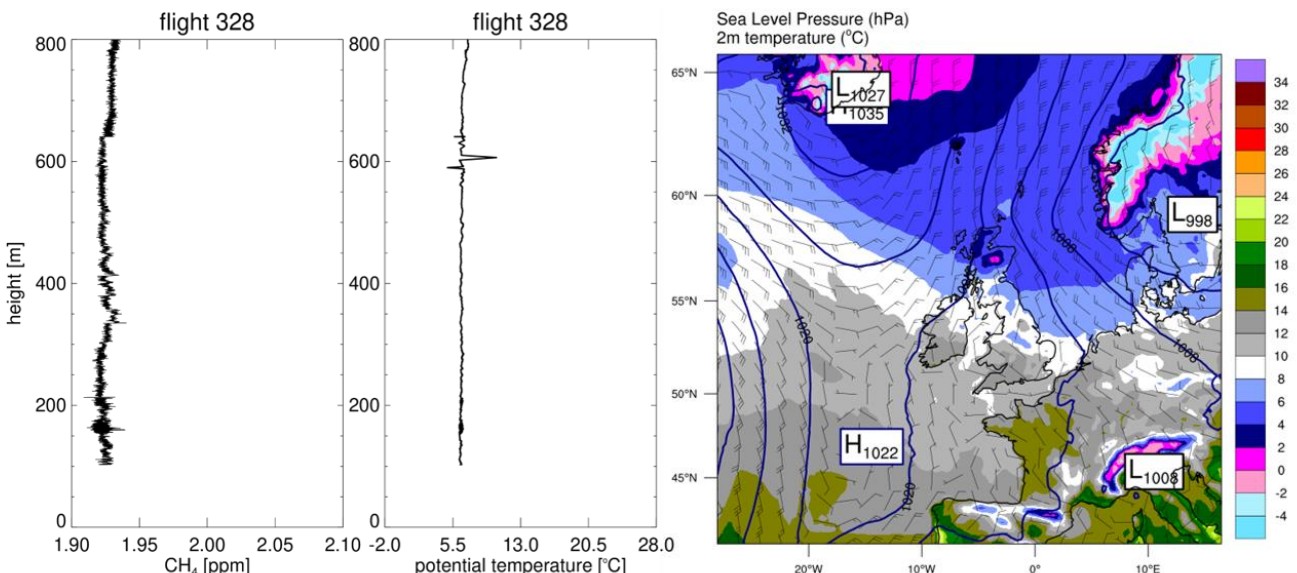

Figure 5. Example of CH₄ and potential temperature profiles in a well-mixed boundary layer under neutral conditions. These are composite profiles flown within an hour over water. The potential temperature and CH₄ profiles stay relatively constant above 200 m. The synoptic chart for 03/05/2019 12:00 UT shows a cyclonic south-easterly air flow over the southern North Sea sector originating from the Benelux.

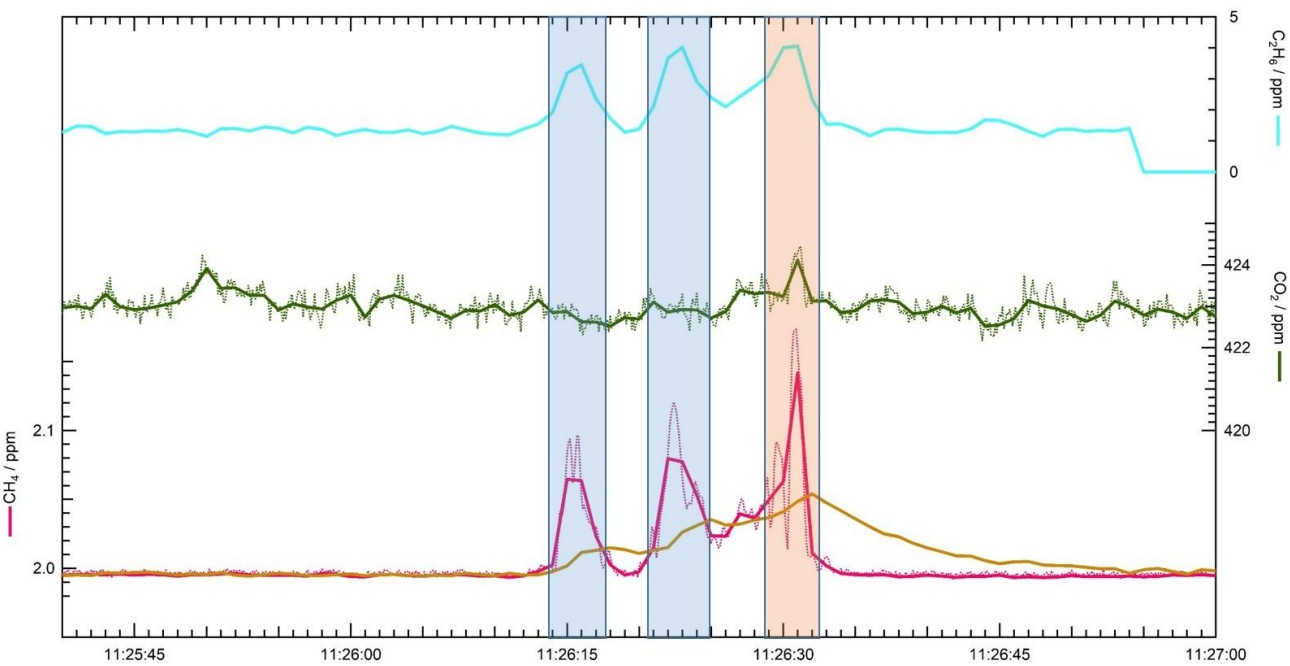


**Figure 6. A cross-section of CH4, CO2 and C2H6 measurement response during one plume sample as recorded by Picarro G2301-f in Pink and Green (10 Hz as dashed lines and downsampled to 1 Hz as solid lines), TILDAS 1 Hz in cyan and Los Gatos uGGA 1 Hz in brown. The difference of the uGGA and Picarro at 1 Hz arises from the slower uGGA response time due to the slower cell turnover. The blue shaded area shows enhancement in C2H6 and CH4, indicating cold venting, the orange shaded area shows**
**enhancement in C2H6 , CH4 and a small amount of CO2 potentially indicating a co-located combustion source.**







**Figure 7. Plumes measured from separate installations to demonstrate the differences in strategies between 2018 and 2019. [upper panel] Plume sampled downwind with poorer vertical spatial resolution in the 2D plane during the 2018 portion of the campaign. CH₄ measured values are much higher due to platform activities during the survey time. [lower panel] Plume sampled downwind in 2019 with intermediate transects enabling higher vertical spatial resolution. Note the colour scale across each plot signifies different measured CH₄; the scales on the upper and lower plots are different.**






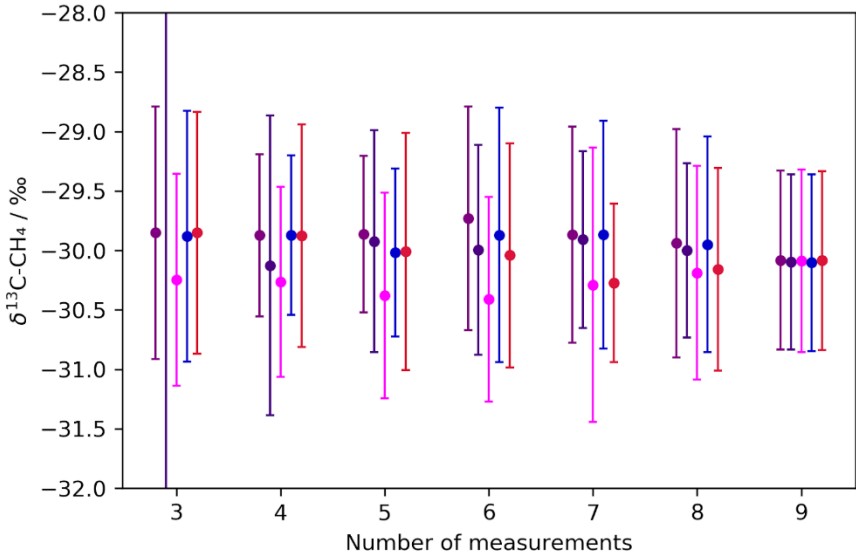

**Figure 8.** $\delta^{13}C_{CH4}$ source signature data source signature data derived from data collected around one installation and assumes a single source of excess methane. The coloured bars represent the uncertainty in the intercept for the regression with varying numbers of sample points removed from the dataset to represent the data if fewer in-plume samples had been collected.

| Survey Year | CH₄ flux lower bound (kT yr⁻¹) | CH₄ flux upper bound (kT yr⁻¹) |
|---|---|---|
| *2018* | 1.83 | 17.9 |
| *2019* | 0.67 | 1.04 |

**Table 1.** A comparison of flux lower and upper bounds for two individual example plumes across each year of survey as scaled by the vertical resolution available. The plumes themselves are not comparable, but the method changes demonstrate the increased certainty in the final results.



|  | Instrument(s) | Method | C2:C1 | Uncertainty |
|---|---|---|---|---|
| *2018 flight* | Los Gatos ultraportable $CH_4/C_2H_6$ | Linear regression | 0.029 | $\pm$ 0.014 |
| *2019 flight* | TILDAS $C_2H_6$ & Picarro G2301-f $CH_4$ | Plume area integration | 0.029 | $\pm$ 0.003 |
| *Published well data* |  |  | 0.031 | $\pm$ 0.009 |

**Table 2. Reported data for C2:C1 for a single installation surveyed during both 2018 and 2019 surveys. Well data from UK oil and gas authority report: https://dataogauthority.blob.core.windows.net/external/DataReleases/ShellExxonMobil/GeochemSNS.zip alongside measured C2:C1 for $CH_4$ enhancements measured during flights in the same geographic area.**



**Author Contribution**

The manuscript was written and figures prepared by JLF, PB and PD with assistance from AEJ, MC, JP, SB and JTS. Experimental design and flight planning was performed by GA, JP, JDL, TLC and DL. Aircraft set-up and in-flight measurements performed by JP, PB, PD, SJA, SY, AIW, TLC, JLF, SW, JW and SB. Laboratory measurements made by REF, RP, SW and SB. Data Processing and calibrations performed by JLF, LH, PB, JTS, PD and MC and SB. Modelling work by NW, JAP, PB and LH.

**Code / Data Availability**

The data for this work will be available via request at the British Antarctic Survey Polar Data Centre.

**Competing interests**

The authors declare that they have no conflict of interest

**Acknowledgements**


This work was funded under the Climate and Clean Air Coalition (CCAC) Oil and Gas Methane Science Studies (MSS), hosted by the United Nations Environment Programme. Funding was provided by the Environmental Defense Fund, Oil and Gas Climate Initiative, European Commission, and CCAC.

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



**Appendices**

**A1. TILDAS data processing and performance**

The TILDAS data was processed as follows. Rapid tuning sweeps of the laser frequency (2996.8 to 2998.0 cm$^{-1}$) by varying the applied current result in the collection of thousands of spectra per second, which are co-averaged. The resulting averaged spectrum is processed at a rate of 1 Hz using a nonlinear least-squares fitting algorithm to determine mixing ratios within the operating software, TDLWintel (© Aerodyne). Averaging of these spectra, and the path length of 76 m achieved using a Herriott multipass cell, provide the sensitivity required for trace gas measurement. Continuously circulated fluid from the

Oasis chiller unit is used as a heat sink for the thermodynamically cooled components and a flow interlock cuts power to the relevant components if the coolant flow stops. Other optical components of the instrument include a 15x Schwarzchild objective in front of each laser, a germanium etalon for measuring the laser tuning rate, a reference gas cell containing air at 25 Torr, and numerous mirrors for adjusting the laser beam alignment. During the airborne campaign the instrument was operated remotely via an Ethernet connection. The TILDAS $C_2H_6$ instrument accuracy has been tested against two standards

containing $C_2H_6$ in mixing ratios of 39.79 ± 0.14 ppb and 2.08 ± 0.02 ppb (high concentration standard and target gas, respectively). As the TILDAS technique relies on highly precise alignment of the focussing and beam-alignment optics before and after the multipass measurement cell it is particularly prone to motion that applies torque to the optical bench. To remove measurement artefacts associated with this sensitivity all data collected for roll angles greater than 20 degrees has been flagged. The presence of the TILDAS in the 2019 campaign ruled out using the multiple circular pass method around a potential

emission source as developed by Scientific Aviation for installation emission flux measurements (Conley et al., 2017) as there was a risk of invalidating data due to the roll angle of the plane if circling tightly around an installation.

**A2. $CO_2$ and $CH_4$ Calibration**

The three cylinders were sampled periodically in-flight to determine the instrument gain factor (slope) and zero-offset for each

analyser. These parameters were linearly interpolated between calibrations and used to rescale the raw measured data (for further details see Pitt et al., 2016). The uncertainties associated with instrument drift and any instrument non-linearity were assessed by sampling the "target" cylinder mid-way between high-low calibrations. The raw target cylinder measurements were rescaled as per the sample data; the mean offset of these target measurements from the WMO-traceable cylinder value (and associated standard deviations) are given for the LGR uGGA and Picarro instrument and are plotted in Figure A4.

The typical duration of calibration cylinder measurements during the 2018 campaign was 45s. The Picarro G2301-f analyser had a high flow rate of ~5 SLPM resulting in rapid flushing of both the inlet tubing and sample cavity. The measured value for each calibration was taken as the average over 15 s prior to the calibration end, as this allowed sufficient time for the measured value to reach equilibrium. The uGGA and uMEA both had much lower flow rates of ~0.5 SLPM, resulting in a much longer equilibration time. Consequently, the calibration duration was not of sufficient length for the uGGA and uMEA



measurements to reach equilibrium and their calibration routine was compromised. For these instruments each calibration run was fitted to an offset exponential function in an attempt to predict the mixing ratio at which equilibration would have occurred, given an infinite amount of calibrating time. In order to improve the data quality and to reduce the post processing time, the calibration periods were run for 75 seconds per cylinder during the 2019 campaign to ensure that all instruments reached equilibrium. Target cylinders were run approximately every 1 hour of flight.

**A3. SWAS Operation**

Each sample is compressed into the canisters using a modified metal bellows pump (Senior Aerospace 28823-7) capable of 150 SLPM open flow but filling the canisters at ~50 SLPM measured average integrated for ~6 and 9 seconds for the 1.4 L and 2 L canisters, respectively. Canister fill pressure is controlled electronically using a back-pressure controller (Alicat, PCR3), BPC. The BPC can maintain flow at any set point pressure (in general 40 psi), including the final fill pressure setpoint.

This allows the 2 L flow through canisters to be filled, even before the operator activates the sampling, enabling air masses to be sampled through which the aircraft has already flown seconds earlier.

Bespoke software was created to allow control of the SWAS system wirelessly from any position in the aircraft using the Ethernet network. Bespoke software was also created for the analysis of the canisters once in the laboratory. The SWAS flown

on the 1st campaign (V1) was a prototype and was updated to the current final version (V2) to fulfil the requirements of the FAAM BAE146 and to address potential issues experienced with the prototype. V2 uses the same canisters and valves as V1 but differs slightly in the size of each case and the plumbing of gas lines. In V2, the canister and valve geometry was optimised to allow an elbow compression fitting between the valve and the canisters to be eliminated, with the valve mounted directly to the canister. This reduces the risk of leaks by 66%. The geometry also allowed the reduction in size by 1U rack unit, allowing

more canisters to be fitted in the same space, improved control electronics and sample logging to ensure canister fill times were captured accurately and stored securely. V2 also saw the addition of 2L flow-through canister cases to complement the 1.4L to-vacuum canister cases. These allowed sample air to be flushed through the canister at a user defined pressure and makes capturing narrow plumes easier due to reduced sample line lag and fill time.






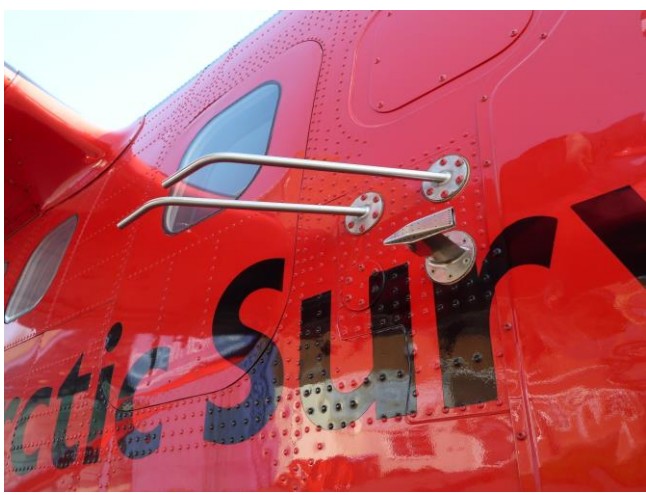

**Figure A1. Photo of the rear-facing chemistry inlets on the BAS Twin Otter aircraft.**

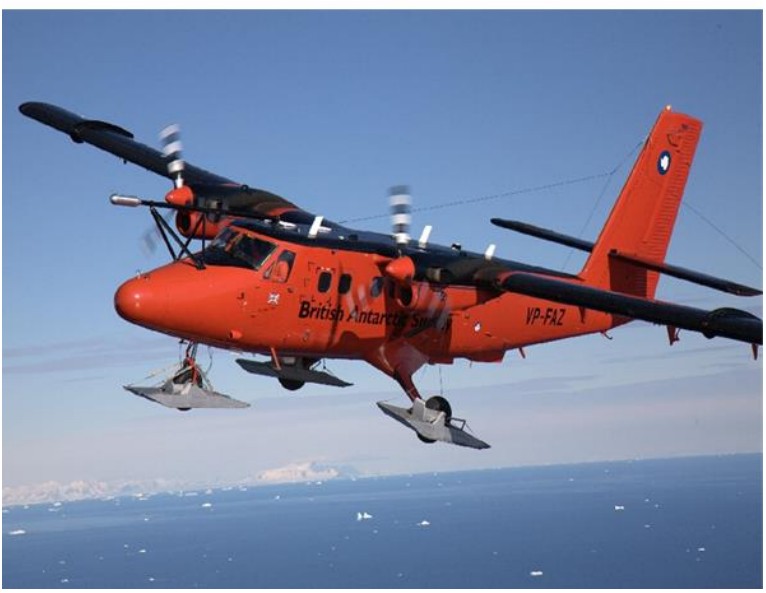

**Figure A2. Photo of the BAS Twin Otter showing the turbulence boom protruding from the front of the aircraft superstructure.**



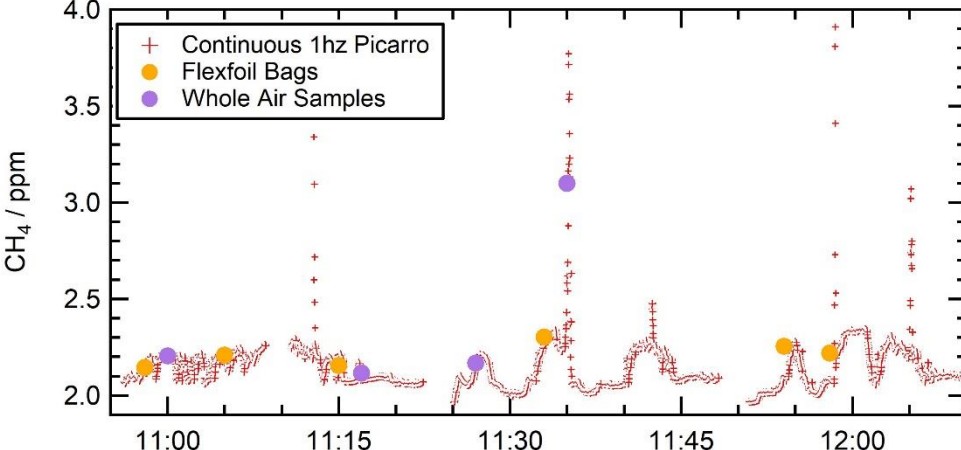


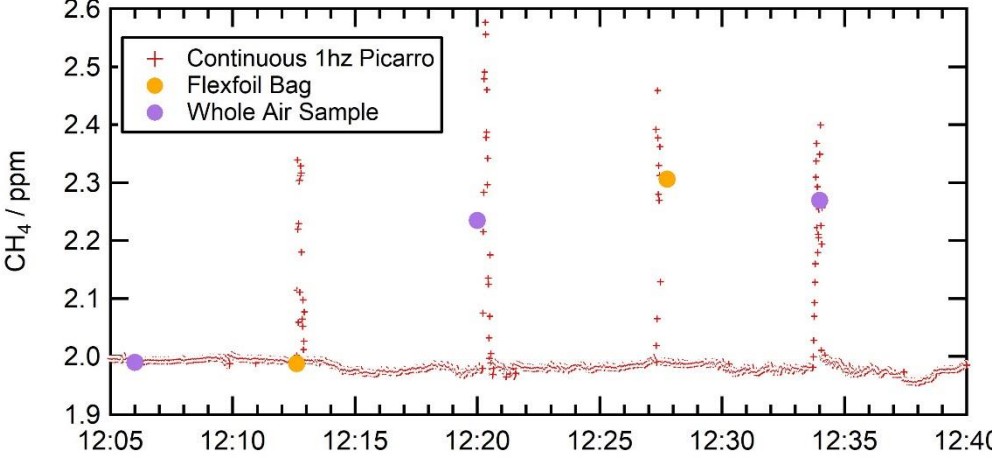

**Figure A3. Examples from a 2018 flight (top panel) and a 2019 flight (lower panel) with attempted capture of CH₄ plumes in spot samples (both SWAS and Flexfoil bags). Note the improved ability to sample at the correct period to capture short-lived enhancement in both SWAS and Flexfoil samples compared for 2019 to 2018 thanks to flight planning and SWAS development improvements.**




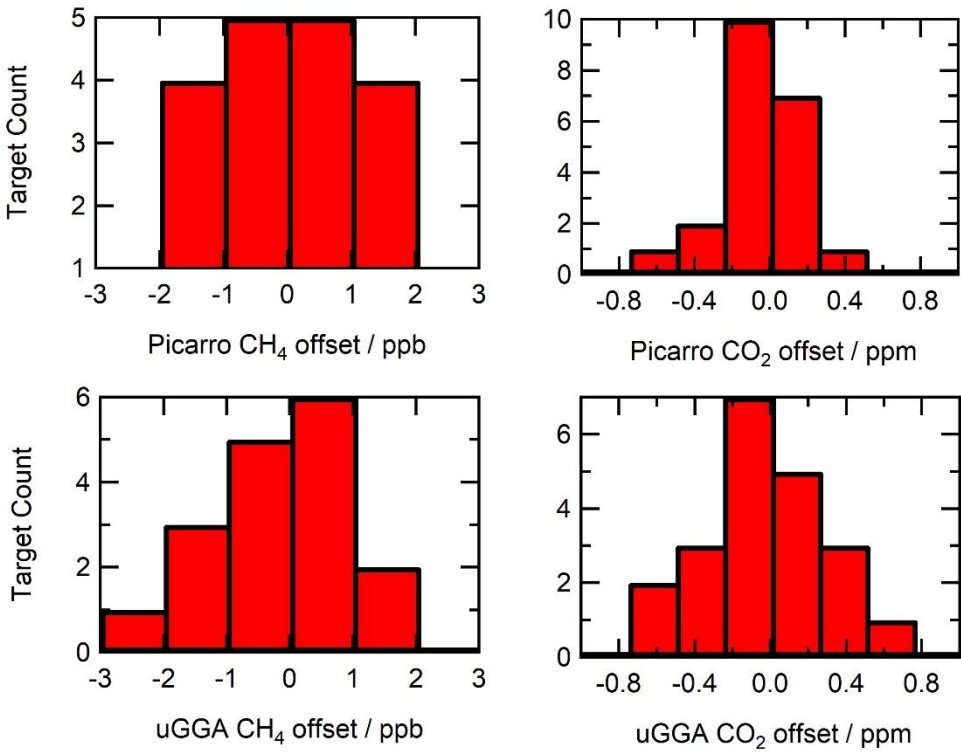

**Figure A4. Target gas data from flights during 2018 for the Picarro G2301-f and Los Gatos uGGA instruments for both CO₂ and CH₄.**




Volatile organic compounds identified and quantified from SWAS samples

| compound | detection limit (ppt) |
| --- | --- |
| ethane | 4 |
| ethene | 4 |
| propane | 6 |
| propene | 2 |
| iso-butane | 1 |
| n-butane | 1 |
| acetylene | 1 |
| trans-2-butene | 2 |
| but-1-ene | 2 |
| cis-2-butene | 2 |
| cyclopentane | 2 |
| iso-butene | 2 |
| iso-pentane | 1 |
| n-pentane | 1 |
| 1,3-butadiene | 2 |
| trans-2-pentene | 2 |
| pent-1-ene | 2 |
| 2,3-methylpentanes | 2 |
| n-hexane | 2 |
| isoprene | 1 |
| n-heptane | 2 |
| benzene | 1 |
| 2,2,4-trimethylpentane | 2 |
| n-octane | 2 |
| toluene | 1 |
| ethylbenzene | 2 |
| m+p- xylenes | 2 |
| o-xylene | 2 |

**Table A1. Summary of VOC's measured from SWAS samples at York University.**