# Peer review of "Facility level measurement of off-shore oil & gas installations from a medium-sized airborne platform: Method development for quantification and source identification of methane emissions"

_Atmospheric Measurement Techniques, 2020_

## Short Comment (SC1) · 8 Jul 2020

I would suggest that you change "small airborne platform" to "medium-sized airborne platform" or similar. Have a look at airborneresearch.org.au to see what "small" (manned airborne platform) means - including for the same type of measurements you are presenting. Apart from that, your paper is great.

Best wishes from the co-developer of the BATprobe you are flying. It probably still

contains our A/D-electronics.

---

## Referee Comment (RC1) · Anonymous Referee #2 · 29 Jul 2020

France et al. quantify CH4 emissions from offshore oil platforms using combinations of instruments aboard a Twin Otter aircraft. They describe the lessons learned from two years of flying downwind of these platforms. They also discuss methods of distinguishing sources of CH4 based on isotopic measurements and correlations with ethane. They find ethane:CH4 emission ratios of 0.029 in both years of flying, in line with published estimates. Their estimates of CH4 mass fluxes improved significantly when flying in 2019 in a well-mixed marine boundary layer.

[Figure]

This paper provides a straightforward description of the project. As such, there is not much to critique. They lessons the authors learned during the two years were mostly to be expected, i.e., faster response instruments were able distinguish source locations better than slow response instruments; a well-mixed marine boundary layer was easier to measure a downwind plume than a layered, poorly-mixed marine boundary layer; etc. However, since the paper will stand as an overview of the project studying emissions from offshore platforms, and because it provides some guidelines for future projects, it is worthy of publication in this journal. I have mostly minor comments related below.

line 54, "pinpoint" seems redundant. Is there a difference between locating and pinpointing emission sources? Maybe the authors mean locate facilities that are emitting, then pinpoint where in the facility the emissions are? And I think this sentence would read better if it were presented in a hypothetical chronological order: first locate emissions, second quantify them, third validate inventories, fourth design effective mitigation.

line 131, stating the precision of the ethane measurement in flight would be more appropriate than in the lab

line 315, when the authors say a "vertical run", do they mean a vertically-stacked horizontal run?

In Figure 7, please state how far downwind the aircraft was for each of these two flights.

line 355, I don't think the word "ideally" is necessary. There must be some variability in the source strength compared to the background in order to fit a line through the data points.

Other comments:

line 253, what does NAME stand for?

Grammar suggestions:
line 23, add comma after (SLCP)

line 93, it looks like the superscript "-1" is a different font

line 103, it looks like the second end parenthesis of the O'Shea reference is a different size?

line 176, it is unclear what "fit" means here

line 181 and elsewhere, suggest changing "in O'Shea" to "by O'Shea"

line 188, suggest "canister sampling" instead of "canisters sampling"

line 199, need ending parenthesis after Lowry reference

line 245, suggest "by Stull" instead of "in Stull"

line 255, change "decision" to "decisions"

line 319, I found this sentence a little confusing to read. I suggest instead of "between the maximal and minimal altitude transects that do not demonstrate CH4 enhancements so are outside of the plume", perhaps say "between the highest and lowest transects without CH4 enhancements, which are above and below the plume, respectively"

line 328, same strange small parenthesis in the Plant reference

line 346, suggest "by Peischl" instead of "in Peischl"

line 352, suggest "by Keeling" instead of "in Keeling"

line 384, "dramatically" is subjective. I suggest removing this word.
* * *

---

## Referee Comment (RC2) · Anonymous Referee #1 · 31 Jul 2020

The Authors describe the development of an airborne measurement platform for the quantification and source attribution of methane from offshore oil and gas operations. The instruments, their airborne deployment and the techniques for data analysis are not really new, but the manuscript would provide a useful reference in future publications that use the data from this platform. I agree that is a worthwhile objective. Overall, the paper is very straightforward and can be published after accounting for the following comments.

[Figure]

Line 56: Suggest "used" instead of "trialled". The latter suggests that previous work should be regarded as somewhat preliminary, but I believe that airborne determination of methane fluxes is quite a mature method by now.

Line 105: A 5-second delay between air entering the inlet and reaching the instruments seems quite long. What is the volume of the sampling manifold and what is the pump speed?

Lines 125-135: I suggest adding a table with the different instruments, parameters measured, measurement precisions and time responses.

Lines 153-154: What mixing ratio would be required for in-flight calibrations and what was available?

Lines 160-165: Is ethane reported as mixing ratios in dry air (as you presumably do for methane)?

Lines 166-174: I re-read papers from two other groups that have used the same TILDAS instrument for airborne measurements of ethane and they seem to have overcome this issue (Smith et al. 2015; Peischl et al. 2018). For example, Smith et al. say that "in-flight drift varied within the instrument precision during a typical research flight" and Peischl et al. gave a "variability of in-flight standard retrieval, $\pm 0.7\%$". These papers should be cited and discussed in this context. How is the airborne deployment of the TILDAS instrument different between the present study and Smith et al. and Peischl et al.? Note that even in a pressurized aircraft, the cabin pressure can still show considerable variability after take-off.

Section 2.5.1 and 2.5.2: When collecting whole air or bag samples in narrow plumes near sources, the exact timing of the sample delay, and open and close times is important to get the best correlation with the in-situ measurements. How well are these known for the instrumentation described here? Fill times are typically a function of altitude.

Lines 220-223: It is not clear if these limitations pertain to the study by Gorchov Negron or to the present manuscript.

Lines 226-238: How far from point sources were the downwind transects typically?

Figures 4 and 5: Perhaps you can add the flight tracks or general location of the flights to the map.

Lines 262-264: Part of the reason that methane shows so little structure in Figure 5 is that it is near the global background. If there are no nearby sources, methane will be perfectly constant regardless of how stable or well-mixed the boundary layer is. Do you have a better example where methane is enhanced more and mixed evenly across the boundary layer?

Section 4.2: Showing that the slow-response instrument is insufficient to separate plumes and determine plume shape is not particularly new or surprising. Does this instrument provide other strengths to justify being part of the payload? For example, is the slow-response instrument more stable and accurate, and allow for important cross calibration opportunities with the fast-response instruments?

Figure 6: Please provide a clearer legend. I found it difficult to decide what is what from the caption and the axis labels.

Equation (1): I found this confusing. Why do you use an average methane enhancement in a plume when you have the time response that allow you to integrate fluxes across a plume (with fluxes in every bin calculated by Eq. 1)? The approach described here relies on a normal distribution of methane in the plumes. Is that true? In addition, this equation yields the flux in units of moles per seconds per meter altitude. This still needs to be integrated across altitude for a meaningful flux number (in moles per second) that can be compared with emissions estimates, but that last step is not included in Equation (1).

Line 323: "vertical extent of the plume" instead of "vertical resolution"?

Lines 340-341: Have you tried to calculate cross correlations between the CH4 and C2H6 measurements to determine the difference in delay times between the two measurements?

Section 5.3: Why not simply show a Keeling plot? The discussion of why such plots are challenging for this application is hard to fully understand without having an example with actual data to look at. Instead, the results are presented in Figure 8 in a very indirect way. The point of this analysis appears to be that with fewer data points (when downwind sampling is less extensive) the deterioration of precision for the 13CH4 delta value is not too severe. But what are the delta values you are trying to distinguish? It might be helpful to add those as horizontal lines in Figure 8 and discuss the loss in precision in terms of those delta values. Overall, the discussion left it unclear to me whether or not the 13CH4 measurements gave useful information. This measurement is one of the more novel aspects of this work and it would be good to see the potential of the method demonstrated in more detail.

Figure 8: What are the differently colored symbols? Also, the caption repeats "source signature data" twice.

Table A1 seems a little out of place as none of these data are used in the manuscript.

References

Peischl, J., Eilerman, S. J., Neuman, J. A., Aikin, K. C., de Gouw, J. A., Gilman, J. B., Herndon, S. C., Nadkarni, R., Trainer, M., Warneke, C. and Ryerson, T. B.: Quantifying methane and ethane emissions to the atmosphere from Central and Western U.S. oil and natural gas production regions, J. Geophys. Res.-Atmos., 123, 7725–7740, doi:10.1029/2018JD028622, 2018.

Smith, M. L., Kort, E. A., Karion, A., Sweeney, C., Herndon, S. C. and Yacovitch, T. I.: Airborne Ethane Observations in the Barnett Shale: Quantification of Ethane Flux and Attribution of Methane Emissions, Environ. Sci. Technol., 49, 8158–8166,

doi:10.1021/acs.est.5b00219, 2015.

---

## Author Comment (AC1) · 23 Oct 2020

We would like to thank the reviewers for their useful and constructive comments regarding the manuscript and have made changes where possible to satisfy their questions and concerns. The original comments from the reviewers are shown in red, with the authors response following in black.

Review comment from Jorg Hacker

I would suggest that you change "small airborne platform" to "medium-sized airborne platform" or similar. Have a look at airborneresearch.org.au to see what "small"(manned airborne platform) means - including for the same type of measurements you are presenting. Apart from that, your paper is great.

Thank you for the positive view of the paper as a whole. The title has been changed to reflect the request made to describe the aircraft as medium sized.

---

## Author Comment (AC2) · 23 Oct 2020

We would like to thank the reviewers for their useful and constructive comments regarding the manuscript and have made changes where possible to satisfy their questions and concerns. The original comments from the reviewers are shown in red, with the authors response following in black.

Anonymous Referee #2

:

France et al. quantify CH4 emissions from offshore oil platforms using combinations of instruments aboard a Twin Otter aircraft. They describe the lessons learned from two years of flying downwind of these platforms. They also discuss methods of distinguishing sources of CH4 based on isotopic measurements and correlations with ethane. They find ethane:CH4 emission ratios of 0.029 in both years of flying, in line with published estimates. Their estimates of CH4 mass fluxes improved significantly when flying in 2019 in a well-mixed marine boundary layer. This paper provides a straightforward description of the project. As such, there is not much to critique. They lessons the authors learned during the two years were mostly to be expected, i.e., faster response instruments were able distinguish source locations better than slow response instruments; a well-mixed marine boundary layer was easier to measure a downwind plume than a layered, poorly-mixed marine boundary layer; etc. However, since the paper will stand as an overview of the project studying emissions from offshore platforms, and because it provides some guidelines for future projects, it is worthy of publication in this journal. I have mostly minor comments related below.

line 54, "pinpoint" seems redundant. Is there a difference between locating and pinpointing emission sources? Maybe the authors mean locate facilities that are emitting, then pinpoint where in the facility the emissions are? And I think this sentence would read better if it were presented in a hypothetical chronological order: first locate emissions, second quantify them, third validate inventories, fourth design effective mitigation.

Agree – this sentence / paragraph has been restructured in line with the suggestions here.

line 131, stating the precision of the ethane measurement in flight would be more appropriate than in the lab

It would indeed be a better metric to provide, however as the cylinders contained no discernible ethane, an in-flight precision measurement estimate is not possible.

line 315, when the authors say a "vertical run", do they mean a vertically-stacked horizontal run?

We agree, this does read better too. "vertical run" has been changed to "stacked horizontal run" through the paper.

In Figure 7, please state how far downwind the aircraft was for each of these two flights.

The distances that the aircraft were for various flights is clarified in the text. The aircraft varied in distance from emitting sources from ~1 to 10km away downwind.

line 355, I don't think the word "ideally" is necessary. There must be some variability in the source strength compared to the background in order to fit a line through the data points.

We agree, this has been changed.

Other comments:

line 253, what does NAME stand for?

The expansion for NAME (Numerical Atmospheric-dispersion Modelling Environment) has been added to the text.

Grammar suggestions:

All subsequent grammar suggestions have been resolved. Thank you for spotting these.

line 23, add comma after (SLCP)

line 93, it looks like the superscript "-1" is a different font

line 103, it looks like the second end parenthesis of the O'Shea reference is a different size?

line 176, it is unclear what "fit" means here

line 181 and elsewhere, suggest changing "in O'Shea" to "by O'Shea"

line 188, suggest "canister sampling" instead of "canisters sampling"

line 199, need ending parenthesis after Lowry reference

line 245, suggest "by Stull" instead of "in Stull"

line 255, change "decision" to "decisions"

line 319, I found this sentence a little confusing to read. I suggest instead of "between

the maximal and minimal altitude transects that do not demonstrate CH4 enhancements

so are outside of the plume", perhaps say "between the highest and lowest

transects without CH4 enhancements, which are above and below the plume, respectively"

line 328, same strange small parenthesis in the Plant reference

line 346, suggest "by Peischl" instead of "in Peischl"

line 352, suggest "by Keeling" instead of "in Keeling"

line 384, "dramatically" is subjective. I suggest removing this word.

**References**

Floerchinger, C., McKain, K., Bonin, T., Peischl, J., Biraud, S. C., Miller, C., Ryerson, T. B., Wofsy, S. C. and Sweeney, C.: Methane emissions from oil and gas production on the North Slope of Alaska, Atmos. Environ., 218, 116985, doi:10.1016/j.atmosenv.2019.116985, 2019.

Gvakharia, A., Kort, E. A., Smith, M. L. and Conley, S.: Testing and evaluation of a new airborne system for continuous N2O, CO2 , CO, and H2O measurements: the Frequent Calibration High-performance Airborne Observation System (FCHAOS), Atmos. Meas. Tech., 11, 6059–6074, doi:10.5194/amt-11-6059-2018, 2018.

Kostinek, J., Roiger, A., Davis, K. J., Sweeney, C., DiGangi, J. P., Choi, Y., Baier, B., Hase, F., Groß, J., Eckl, M., Klausner, T. and Butz, A.: Adaptation and performance assessment of a quantum and interband cascade laser spectrometer for simultaneous airborne in situ observation of CH4, C2H6, CO2, CO and N2O, Atmos. Meas. Tech., 12, 1767–1783, doi:10.5194/amt-12-1767-2019, 2019.

Peischl, J., Eilerman, S. J., Neuman, J. A., Aikin, K. C., de Gouw, J., Gilman, J. B., Herndon, S. C., Nadkarni, R., Trainer, M., Warneke, C. and Ryerson, T. B.: Quantifying Methane and Ethane Emissions to the Atmosphere From Central and Western U.S. Oil and Natural Gas Production Regions, J. Geophys. Res. Atmos., 123, 7725–7740, doi:10.1029/2018JD028622, 2018.

Pitt, J. R., Le Breton, M., Allen, G., Percival, C. J., Gallagher, M. W., Bauguitte, S. J.-B., O'Shea, S. J., Muller, J. B. A., Zahniser, M. S., Pyle, J. and Palmer, P. I.: The development and evaluation of airborne in situ N2O and CH4 sampling using a quantum cascade laser absorption spectrometer (QCLAS), Atmos. Meas. Tech., 9, 63–77, doi:10.5194/amt-9-63-2016, 2016.

Santoni, G. W., Daube, B. C., Kort, E. A., Jiménez, R., Park, S., Pittman, J. V., Gottlieb, E., Xiang, B., Zahniser, M. S., Nelson, D. D., McManus, J. B., Peischl, J., Ryerson, T. B., Holloway, J. S., Andrews, A. E., Sweeney, C., Hall, B., Hintsa, E. J., Moore, F. L., Elkins, J. W., Hurst, D. F., Stephens, B. B., Bent, J. and Wofsy, S. C.: Evaluation of the airborne quantum cascade laser spectrometer (QCLS) measurements of the carbon and greenhouse gas suite – CO2, CH4, N2O, and CO – during the CalNex and HIPPO campaigns, Atmos. Meas. Tech., 7, 1509–1526, doi:10.5194/amt-7-1509-2014, 2014.

Smith, M. L., Kort, E. A., Karion, A., Sweeney, C., Herndon, S. C. and Yacovitch, T. I.: Airborne Ethane Observations in the Barnett Shale: Quantification of Ethane Flux and Attribution of Methane Emissions, Environ. Sci. Technol., 49, 8158–8166, doi:10.1021/acs.est.5b00219, 2015.

---

## Author Comment (AC3) · 23 Oct 2020

We would like to thank the reviewers for their useful and constructive comments regarding the manuscript and have made changes where possible to satisfy their questions and concerns. The original comments from the reviewers are shown in red, with the authors response following in black.

Anonymous Referee #1

The Authors describe the development of an airborne measurement platform for the quantification and source attribution of methane from offshore oil and gas operations. The instruments, their airborne deployment and the techniques for data analysis are not really new, but the manuscript would provide a useful reference in future publications that use the data from this platform. I agree that is a worthwhile objective. Overall, the paper is very straightforward and can be published after accounting for the following comments.

Line 56: Suggest "used" instead of "trialled". The latter suggests that previous work should be regarded as somewhat preliminary, but I believe that airborne determination of methane fluxes is quite a mature method by now.

We Agree – this has now been changed in the manuscript.

Line 105: A 5-second delay between air entering the inlet and reaching the instruments seems quite long. What is the volume of the sampling manifold and what is the pump speed?

The 5 second delay was measured for the uGGA by spiking the aircraft inlet tip using breath $CO_2$, and monitoring the concentration response of the spectrometer. The lagtime was also confirmed by an analysis of the various inlet section showed in Figure 2, their dead volumes and plug-flow flush rates, namely: 36 $cm^3$ for the 52 cm long ½" od SS rearward facing inlet flushed at ~15000 $cm^3$/min (inlet pump), and 16 $cm^3$ for the 110 cm long ¼" od Synflex transfer line flushed at ~ 200 $cm^3$/min (uGGA internal pump), combining to ~5 sec.

For the Picarro spectrometer, which external pump maintains a sample flowrate of 5000 $cm^3$/min, the combined lagtimes of its three inlet sections can be estimated at 0.14 + 0.14 + 0.16 ~0.4 sec.

We admit that our quoted 5 sec is an overestimate for the Picarro CRDS. As the lagtime varies for each instrument, this sentence has been removed as it is not especially helpful to the reader.

Lines 125-135: I suggest adding a table with the different instruments, parameters measured, measurement precisions and time responses.

A table with measured precisions, response and lag times has been collated for the Appendix and replaced some of the description in the text.

Lines 153-154: What mixing ratio would be required for in-flight calibrations and what was available?

There was no discernible ethane in the cylinder mixes and therefore not possible to complete in-flight calibrations. This is now clarified in the manuscript text.

Lines 160-165: Is ethane reported as mixing ratios in dry air (as you presumably do for methane)?

The TILDAS outputs the dry mixing ratio as it monitors a water line itself that is used to calculate, and correct the mixing ratio. The TILDAS ethane measurement description has this information added to the manuscript.

Lines 166-174: I re-read papers from two other groups that have used the same TILDAS instrument for airborne measurements of ethane and they seem to have overcome this issue (Smith et al. 2015; Peischl et al. 2018). For example, Smith et al. say that "in-flight drift varied within the instrument precision during a typical research flight" and Peischl et al. gave a "variability of in-flight standard retrieval, ±0.7%". These papers should be cited and discussed in this context. How is the airborne deployment of the TILDAS instrument different between the present study and Smith et al. and Peischl et al.? Note that even in a pressurized aircraft, the cabin pressure can still show considerable variability after take-off.

It is true that there are several published papers containing data from TILDAS instruments. However, the two papers cited here by the reviewer focus on the estimated fluxes rather than a detailed analysis of instrument performance. Therefore, we don't believe it would be appropriate to cite them as evidence that the cabin pressure sensitivity has been overcome in these cases, because such evidence is not provided. The relevance of the in-flight standard variability from Peischl et al. (2018) to this issue is not clear, because the paper does not state over what range of altitudes these standard measurements were taken. The study by Smith et al. (2015) states that the instrument stability was evaluated based on the difference  between free troposphere measurements at the beginning and end of a flight. This accounts for temporal drift but not necessarily sensitivity to cabin pressure changes (depending on whether the data comparison was conducted over the same altitude range).

To our knowledge, there are four instrument-focussed papers reporting the performance of Aerodyne TILDAS instruments on aircraft (Santoni et al., 2014; Pitt et al., 2016; Gvakharia et al., 2018; Kostinek et al., 2019). Of these, the three most recent papers all report measurement sensitivity to cabin pressure. Gvakharia et al. (2018) and Kostinek et al. (2019) use frequent measurements of zero air and/or calibration gas at very high flow rates to correct for this issue while maintaining a reasonably high duty cycle. These measurements are performed every 2 minutes by Gvakharia et al. (2018) and every 5-10 minutes by Kostinek et al. (2019). Both studies demonstrate that at these frequencies even the rapid drift during vertical profiles can be adequately accounted for. On the other hand, Floerchinger et al. (2019) made zero measurements every 10 minutes, and still observed a strong sensitivity to cabin pressure changes (C. Floerchinger, personal communication). This wasn't reported in the manuscript because the vertical profile data was not required for the study. Peischl et al. (2018) and Smith et al. (2015) made zero measurements every 15 minutes but do not provide data on stability during vertical profiles (again, it doesn't appear that this data was used in the mass balance flux calculations).

Santoni et al. (2014) do not explicitly demonstrate the measurement sensitivity (or lack thereof) to cabin pressure changes. One of the two instruments they describe (the QCLS-CO2) is housed within a hermetically sealed pressure vessel, which presumably solves this issue. Developing a similar customised hardware fix would be one option to improve performance on the BAS Twin Otter going forward. Alternatively, if payload constraints allow, it may prove simpler to implement a frequent, fast calibration system of the sort used by Gvakharia et al. (2018) and Kostinek et al. (2019). However, as discussed in the manuscript, as long as each in-plume measurement is referenced against a background measurement at the same altitude the calculated enhancements are not impacted by this cabin

pressure sensitivity. Therefore, this issue does not have a serious impact on the results presented in this study. We have added references to Santoni et al. (2014) and Kostinek et al. (2019) to the manuscript.

Section 2.5.1 and 2.5.2: When collecting whole air or bag samples in narrow plumes near sources, the exact timing of the sample delay, and open and close times is important to get the best correlation with the in-situ measurements. How well are these known for the instrumentation described here? Fill times are typically a function of altitude.

The fill times are very rapid (of the order of a few seconds) with this configuration for the bag samples and the Whole Air Sample collection benefitted from the upgrade to the continuous through-flow for the second campaign. As samples are taken at low altitude passes within the boundary layer, the problem of altitude adjusted fill times was not a problem, although this is a noted problem on high altitude flying on aircraft such as FAAM (France, Cain et al. 2016).

Lines 220-223: It is not clear if these limitations pertain to the study by Gorchov Negron or to the present manuscript.

This is clarified in the text.

Lines 226-238: How far from point sources were the downwind transects typically?

Transects varied between 1 and 10 km downwind of the source rig locations. In some cases, for example where multiple potential sources were in the same upwind direction, the maximum distance downwind may have been greater than 10 km – analysis of the attribution of point sources will be dealt with in future work. This has been added to the text.

Figures 4 and 5: Perhaps you can add the flight tracks or general location of the flights to the map.

General non-specific area locations have been added to the figures

Lines 262-264: Part of the reason that methane shows so little structure in Figure 5 is that it is near the global background. If there are no nearby sources, methane will be perfectly constant regardless of how stable or well-mixed the boundary layer is. Do you have a better example where methane is enhanced more and mixed evenly across the boundary layer?

A new example from the available data has been chosen to reflect this more clearly in Figure 5.

Section 4.2: Showing that the slow-response instrument is insufficient to separate plumes and determine plume shape is not particularly new or surprising. Does this instrument provide other strengths to justify being part of the payload? For example, is the slow-response instrument more stable and accurate, and allow for important cross calibration opportunities with the fast-response instruments?

It serves as a relatively lightweight and cost effective redundancy in this configuration. Although with minor modification to the set-up it could have been run in a faster mode, the LGR uGGA would still not have matched the fast Picarro instrument for performance. It was therefore kept as a simple tool for cross-checking and redundancy. The lack of detail in the plume measured by the slower instrument can be clearly seen in Figure 6, but it still serves as a useful cross-checking dataset.

Figure 6: Please provide a clearer legend. I found it difficult to decide what is what from the caption and the axis labels.

Figure 6 has been updated with a new set of legends for the figure.

Equation (1): I found this confusing. Why do you use an average methane enhancement in a plume when you have the time response that allow you to integrate fluxes across a plume (with fluxes in every bin calculated by Eq. 1)? The approach described here relies on a normal distribution of methane in the plumes. Is that true? In addition, this equation yields the flux in units of moles per seconds per meter altitude. This still needs to be integrated across altitude for a meaningful flux number (in moles per second) that can be compared with emissions estimates, but that last step is not included in Equation (1).

The equation employed for mass balance here does use average values for methane evaluated across the horizontal width of the plume. However, as this mean is evaluated over an explicit plume width ($\Delta x$), the flux calculated is exactly equal to a piecewise (i.e. smaller dx) integration over discrete horizontal distances. There is no requirement for the plume to be Gaussian, or for a normal distribution of measured concentrations when in the plume. The only assumption being made here is that the aircraft speed and heading remain constant as it passes through the plume – a good assumption in the case of narrow single-facility plumes like the ones presented here.

Also, the reviewer is correct that the equation is missing the vertical height of the plume. The equation has been amended to include an additional "$\Delta z$" term, relating to the vertical extent of the plume.

Line 323: "vertical extent of the plume" instead of "vertical resolution"?

 We agree, this reads better as "vertical extent of the plume".

Lines 340-341: Have you tried to calculate cross correlations between the CH4 and C2H6 measurements to determine the difference in delay times between the two measurements?

This was attempted, but the results were not suitable due to the variability in the different turnover time (e-fold response) of the instruments and significantly affected the ratios. We expect the integration method to be more accurate.

Section 5.3: Why not simply show a Keeling plot? The discussion of why such plots are challenging for this application is hard to fully understand without having an example with actual data to look at. Instead, the results are presented in Figure 8 in a very indirect way. The point of this analysis appears to be that with fewer data points (when downwind sampling is less extensive) the deterioration of precision for the 13CH4 delta value is not too severe. But what are the delta values you are trying to distinguish? It might be helpful to add those as horizontal lines in Figure 8 and discuss the loss in precision in terms of those delta values. Overall, the discussion left it unclear to me whether or not the 13CH4 measurements gave useful information. This measurement is one of the more novel aspects of this work and it would be good to see the potential of the method demonstrated in more detail.

A Keeling plot is now shown along with the current Figure 8. We agree that this was a bit of an oversight not to include this data plotted in this way.

Figure 8: What are the differently colored symbols? Also, the caption repeats "source signature data" twice.

Caption has been updated to correct this and clarify.

Table A1 seems a little out of place as none of these data are used in the manuscript.

Although we agree, we would like to keep this in as it is a demonstration of measurement capability and fits with the theme of the paper.

**References**

Floerchinger, C., McKain, K., Bonin, T., Peischl, J., Biraud, S. C., Miller, C., Ryerson, T. B., Wofsy, S. C. and Sweeney, C.: Methane emissions from oil and gas production on the North Slope of Alaska, Atmos. Environ., 218, 116985, doi:10.1016/j.atmosenv.2019.116985, 2019.

France, J. L., M. Cain, R. E. Fisher, D. Lowry, G. Allen, S. J. O'Shea, S. Illingworth, J. Pyle, N. Warwick, B. T. Jones, M. W. Gallagher, K. Bower, M. Le Breton, C. Percival, J. Muller, A. Welpott, S. Bauguitte, C. George, G. D. Hayman, A. J. Manning, C. L. Myhre, M. Lanoisellé and E. G. Nisbet (2016). Measurements of δ13C in CH4 and using particle dispersion modeling to characterize sources of Arctic methane within an air mass. Journal of Geophysical Research: Atmospheres 121(23): 14,257-214,270.

Gvakharia, A., Kort, E. A., Smith, M. L. and Conley, S.: Testing and evaluation of a new airborne system for continuous N2O, CO2 , CO, and H2O measurements: the Frequent Calibration High-performance Airborne Observation System (FCHAOS), Atmos. Meas. Tech., 11, 6059–6074, doi:10.5194/amt-11-6059-2018, 2018.

Kostinek, J., Roiger, A., Davis, K. J., Sweeney, C., DiGangi, J. P., Choi, Y., Baier, B., Hase, F., Groß, J., Eckl, M., Klausner, T. and Butz, A.: Adaptation and performance assessment of a quantum and interband cascade laser spectrometer for simultaneous airborne in situ observation of CH4, C2H6, CO2, CO and N2O, Atmos. Meas. Tech., 12, 1767–1783, doi:10.5194/amt-12-1767-2019, 2019.

Peischl, J., Eilerman, S. J., Neuman, J. A., Aikin, K. C., de Gouw, J., Gilman, J. B., Herndon, S. C., Nadkarni, R., Trainer, M., Warneke, C. and Ryerson, T. B.: Quantifying Methane and Ethane Emissions to the Atmosphere From Central and Western U.S. Oil and Natural Gas Production Regions, J. Geophys. Res. Atmos., 123, 7725–7740, doi:10.1029/2018JD028622, 2018.

Pitt, J. R., Le Breton, M., Allen, G., Percival, C. J., Gallagher, M. W., Bauguitte, S. J.-B., O'Shea, S. J., Muller, J. B. A., Zahniser, M. S., Pyle, J. and Palmer, P. I.: The development and evaluation of airborne in situ N2O and CH4 sampling using a quantum cascade laser absorption spectrometer (QCLAS), Atmos. Meas. Tech., 9, 63–77, doi:10.5194/amt-9-63-2016, 2016.

Santoni, G. W., Daube, B. C., Kort, E. A., Jiménez, R., Park, S., Pittman, J. V., Gottlieb, E., Xiang, B., Zahniser, M. S., Nelson, D. D., McManus, J. B., Peischl, J., Ryerson, T. B., Holloway, J. S., Andrews, A. E., Sweeney, C., Hall, B., Hintsa, E. J., Moore, F. L., Elkins, J. W., Hurst, D. F., Stephens, B. B., Bent, J. and Wofsy, S. C.: Evaluation of the airborne quantum cascade laser spectrometer (QCLS) measurements of the carbon and greenhouse gas suite – CO2, CH4, N2O, and CO – during the CalNex and HIPPO campaigns, Atmos. Meas. Tech., 7, 1509–1526, doi:10.5194/amt-7-1509-2014, 2014.

Smith, M. L., Kort, E. A., Karion, A., Sweeney, C., Herndon, S. C. and Yacovitch, T. I.: Airborne Ethane Observations in the Barnett Shale: Quantification of Ethane Flux and Attribution of Methane Emissions, Environ. Sci. Technol., 49, 8158–8166, doi:10.1021/acs.est.5b00219, 2015.